# Decellularized Human Dermis for Orthoplastic Extremity Reconstruction

**DOI:** 10.3390/bioengineering11050422

**Published:** 2024-04-25

**Authors:** Christopher Bibbo, K. Ümit Yüksel

**Affiliations:** 1Rubin Institute for Advanced Orthopaedics, International Center for Limb Lengthening, Sinai Hospital of Baltimore, 2401 West Belvedere Avenue, Baltimore, MD 21215, USA; 2Independent Scientific Researcher, Kennesaw, GA 30144, USA

**Keywords:** wound healing, ECM, decellularization, dermis, dCELL, DermaPure, orthoplastic, extremity reconstruction, tendon, wounds, ligament, skin graft

## Abstract

The reconstruction of patients who possess multi morbid medical histories remains a challenge. With the ever-increasing number of patients with diabetes, infections, and trauma, there is a consistent need for promotion of soft tissue healing and a reliable substrate to assist with every aspect of soft tissue reconstruction, as well as the loss of fascial domain. Several proprietary products filled some of these needs but have failed to fulfill the needs of the clinician when faced with reconstructing multiple soft tissue systems, such as the integument and the musculoskeletal system. In this paper we discuss the use of decellularized human dermis (DermaPure^®^, Tissue Regenix, Universal City, TX, USA) through which a unique human tissue processing technique (dCELL^®^ technology, Tissue Regenix, Universal City, TX, USA) and the creation of multiple product forms have proven to exhibit versatility in a wide range of clinical needs for successful soft tissue reconstruction. The background of human tissue processing, basic science, and early clinical studies are detailed, which has translated to the rationale for the success of this unique soft tissue substrate in orthoplastic reconstruction, which is also provided here in detail.

## 1. Introduction

The aim of this review article is to discuss decellularized tissue engineering, with specific emphasis on decellularized human dermis development, characterization, and clinical outcomes.

The term tissue engineering conjures a picture where tissue is constructed de novo from its components. While efforts in that direction have been made, and totally synthetic non-viable grafts have been on the market for decades, tissue engineering lies in transforming donated human tissue into a form that is suitable for transplantation into another human.

### 1.1. Origins of Allograft Tissue

In his 1957 paper, Hyatt recounts the origin of the Navy Tissue Bank in 1949 as a deep freezer for bone storage, its progress, and traces the origins to the development of following physical principles as applied to storage of human tissue: freezing and storage at −20 °C, refrigerated storage (4 °C) of nutrient media, rapid “protective” freezing to −76 °C, and freeze-drying and storage [1]. He indicates the Korean conflict highlighted some deficiencies in reparative surgery, with available tissue being hampered by both procurement (quality of tissue and excision immediately postmortem) and preservation techniques. He points out that nutrient cold storage at 4 °C affords viability up to 6 weeks; maintaining viability by soaking tissue in glycerol-saline and freezing at dry ice temperatures (−76 °C), freeze-frying of tissue after slow-freezing of bone, or rapid-freezing of soft tissue which eliminates viability but maintains shape and size affords long term storage and reconstitution. He reports the success of the program as measured by the ability to treat fractures or serious defects with stored bone, and ability to supply large amounts of skin for use in burn victims.

The origins of US Navy Tissue Bank, along with its contributions to the development of tissue banks including the National Marrow Donor Program and the American Association of Tissue Banks (AATB) in the USA, and other notable achievements, including the identification of appropriate donor criteria for tissue donation, the development of procurement and processing methods, the establishment of a graft registry, and documentation and the clinical evaluation of a variety of tissues, cryopreservation, freeze-drying, irradiation sterilization of tissue, and immunological principles of tissue transplantation, are chronicled in an article by Strong, published on the occasion of the 50th anniversary of the US Navy Tissue Bank [2].

### 1.2. Early Developments in Tissue Processing and Addressing the Immune Response

In his address at the First Annual Meeting of the Society for Cryobiology, Gresham discusses the method for cryopreserving tissue and how it evolved in a short time span of about a year [3]. He also discusses the case for different tissues, noting “It was only natural that its application should be directed toward bone banking, since the clinical use of nonviable bone homografts was well established …” indicating the choice of tissue to be freeze-dried was not random. His discussion is weighted on bone, since that was the most preserved and utilized tissue at that time. Based on the limited amount of clinical data, he quotes an 85% success rate for preserved allografts and finds that comparable to that of autografts (93%). It is interesting to note that he discusses the introduction of bone particles to the field, and that these are as well tolerated as the intact bone pieces. In the case of fascia, he speculates that it and other collagen grafts, such as tendon and dura, function directly according to their application and do not necessarily have to be replaced by the host. In case of skin grafts used in severely burned patients, he indicates that a critically significant statistical study is hampered by a multitude of factors leading to the clinical outcome of these cases. He reports viable or nonviable homografts consistently remain attached to the host for a relatively short period (10–14 days), and in contrast to fresh autografts merely function as a nonviable passively adherent biological membrane, and that freeze-drying serves primarily as a means for long-term storage. He points out the unique histological structure of arteries and their elasticity, that it needs to be maintained for proper long-term function, and improper freezing can alter the elastic property.

Abbott and Pappas report on fundamental biologic differences in the behavior of fresh and preserved skin as grafts in experiments conducted in mice [4]. They note that the goal in burn care is autograft coverage, the optimum is a bed for the autograft in the shortest time and an optimum granulating site is produced by using the allograft as a biologic dressing. They compare skin stored by four different methods: (1) viable fresh; (2) viable frozen (soaked in 15% dimethylsulfoxide (DMSO)–Hank’s balanced salt solution, then stored frozen at −156 °C in liquid nitrogen vapor); (3) non-viable frozen (−70 °C. freezer); and (4) non-viable freeze dried (frozen at −70 °C, then freeze-dried in a vacuum chamber). There were differences in both the preparation steps before storage and the reconstitution steps prior to implantation. They made no effort to correlate storage time (10 days to 11 months) or details of storage methods with the results. In comparing the behavior of isografts and allografts processed with above methods they found that isografts of groups 1 and 2 behaved as autografts and survived permanently, while those of groups 3 and 4 were rejected as were similarly treated allografts (*p* < 0.01). They suggested that rejection of nonviable grafts may not be influenced by factors other than the immune responses. They found rejection of freeze-dried allografts is more rapid in animals with smaller wounds than in those with larger wounds. Based upon this finding, they proposed that survival of freeze-dried grafts should not be influenced by specific hyperimmunity against second set skin grafts as would fresh skin, and both tested and proved this hypothesis.

In addition to some of the technological developments listed by Hyatt, Sell—along with Perry and Stevenson—is credited for introducing surface antigen typing to tissue banking [5]. These efforts led to the hypothesis that certain antibodies and ligands can be used to induce long-term allograft survival. This hypothesis was tested in a preclinical primate (rhesus monkeys) model, using CTLA4-Ig and the CD40L-specific monoclonal antibody 5C8. While control animals rejected the transplant in 5–8 days, interaction of either of these agents prolonged rejection-free survival (20–98 days), and 2/4 animals treated with both agents experienced extended (>150 days) rejection-free allograft survival. They concluded that CTLA4-Ig and 5C8 can both prevent and reverse acute allograft rejection, significantly prolonging the survival of major histocompatibility complex-mismatched renal allografts in primates without the need for chronic immunosuppression [6].

Attempts have been made to “hide” the immunogenic component by crosslinking the tissue with glutaraldehyde or carbodiimide. An example is the CryoLife-O’Brien stentless porcine aortic valve. Good results with this valve have been reported in 240 patients, with 97% of the patients exhibiting zero or a trace incompetence at 2 1/2 years [7]. An issue with tissue that has been tanned, i.e., fixed or crosslinked, with glutaraldehyde is calcification of the implanted tissue over time. Use of a sulfated higher aliphatic alcohol such as sodium dodecyl sulfate (SDS) has been proposed to mitigate this reaction [8]. Another issue with glutaraldehyde crosslinking includes stiffening of tissue due to the short chain of the dialdehyde molecule as well as by the degree of crosslinking. The short chain of the dialdehyde molecule creates short and stiff bons. In addition, the short chain may react only with one end since the other end may not reach another lysine residue. Processors include neutralization steps such as washing the processed tissue with glycine to neutralize the remaining reactive moiety. Mathapati et al. proposed detoxification of glutaraldehyde crosslinked bovine pericardium by decellularization, pretreating it with ethanol or removing the free aldehydes by citric acid treatment and lyophilization [9]. They report no significant change in biomechanical properties and biodegradability when enzymatic hydrolysis (*p* > 0.05) after this treatment compared with the native detoxified glutaraldehyde crosslinked bovine pericardium. On the other hand, a statistically significant (*p* < 0.05) increase in the viability of endothelial progenitor cells (EPCs) and cell adhesion after the proposed treatment was observed.

### 1.3. Decellularization of Tissue

Except for the crosslinked tissue discussed above, it was thought that the viability of soft tissues (specifically heart valves and veins) like those of organs was essential for their function. These tissues were kept either cooled or cryopreserved, and then implanted. In the 1990s it became evident that, once implanted, these tissues quickly lost their viable cells, and shed their endothelium where applicable; consequently, it took quite a while to clear the tissue from donor cells and for the recipient’s cells to colonize them. In other words, remodeling was slow, and benefits were delayed. With this realization, the thought process started to shift. Soon, it was clear that tissue decellularized in a manner that did not affect the structure of the collagen matrix was equally effective, if not better, in remodeling. The recipient’s own cells more readily populated the decellularized matrices. The utilization decellularized grafts was proposed to diminish their immunogenicity and thereby delaying their degeneration. Several companies have developed processes to decellularize heart valves, veins, and arteries for transplanting from one person to another (allografts). Elkins et al. reported on reduced antigenicity of heart valves using the SynerGraft^®^ technology [10]. SynerGraft treatment reduced tissue antigen expression without altering human valve biomechanics or strength. CryoValve^®^ SG, human pulmonary valves treated with SynerGraft technology, did not provoke a panel reactive antibody response when implanted in humans. When evaluated in sheep, recellularization with the host tissue was observed. In another study, the function and durability of SynerGraft-treated heart valves in sheep were assessed [11]. Recellularization was evaluated histologically through 11 months, with cell phenotypes identified using specific antibodies. The authors reported that these heart valves were progressively recellularized, starting at the conduit, then in the leaflet, and is associated with revascularization of the graft, and functional, fully developed fibrocytes, actively synthesizing type I procollagen, were present within 3 months. After 11 months, leaflet explants had no detectable inflammatory cells, were as much as 80% repopulated, and had a distribution of smooth muscle actin positive cells resembling that of the natural leaflet.

Decellularization was also studied at the Pontifical Catholic University of Paraná (PUCPR), and da Costa and colleagues published on this work. Their process consisted of immersing the valves for 24 h in RPMI nutrient medium containing antibiotics, soaking tissue in sodium dodecyl sulphate (SDS) solution for ≥10 days, and maintaining the tissue in RPMI nutrient medium with antibiotics at 4 °C until implanted. This was evaluated in eight juvenile sheep orthotopically implanted with decellularized valves through echocardiograms and histological analysis via hematoxylin-eosin, Movat’s, and alizarin-red staining of explants. At the predetermined 90- and 180-day termination points, animals had significant somatic growth and the implants had increased diameter without central valve insufficiency. Histologically, all homografts preserved their extracellular matrix organization and were progressively recellularized without calcification. A shortcoming of this study is the small number of subjects enrolled in this study [12]. Preclinical studies of decellularized heart valves in sheep demonstrated the value of this technology. Both in vitro biomechanical and hydrodynamic tests and in vivo biocompatibility tests proved the technology both very valuable and successful. The technology was then used to implant decellularized pig valves in humans. A retrospective clinical study on 129 patients was published showing both the utility and efficacy of such decellularized valves in humans over early- and mid-term follow-up results [13]. Da Costa later reported on the long-term results of the Ross operation, involving four groups of patients depending on the type of the allograft used for the right ventricle outflow tract (RVOT) reconstruction: cryopreserved allografts (Group 1, n = 204), cryopreserved allografts decellularized with deoxycholic acid (Group 2, n = 44), cryopreserved allografts decellularized with a SDS-based solution (Group 3, n = 42), and fresh allografts decellularized with SDS (Group 4, n = 124). At 6 years, fresh allografts decellularized with SDS were significantly free of structural dysfunction when compared with the other groups [14]. Valved patches made from decellularized human homograft were used to enlarge stenosis in patients with tetralogy of Fallot (TOF). The mean follow-up duration for the 15 patients with TF who needed enlargement of the right ventricular outflow tract (RVOT) was 12.7 months (1–25 months). They concluded that the medium-term follow-up results suggested the decellularized allograft seemed to be a good option for ROVT enlargement in patients with TOF [15].

Melandri and colleagues also developed a decellularization technology [16]. Their method is based on enzymatic digestion of the tissue, preferably with trypsin for 12–24 h at about 37 °C; it does not use the detergents used by other methods. Their evaluation indicated highly efficient decellularization based on residual DNA, GAG and collagen content measurements, vitality index, histology, and electron microscopy [17]. In a later case report, they showed successful use of meshed decellularized dermal matrix overlaid with an autologous split thickness skin graft on a burn injury on the dorsal side of the foot [18].

Xu et al. demonstrated the importance of removing the galactose-α-(1,3)-galactose terminal disaccharide (α-Gal) for host response, and that cellular remnants in human dermis grafts implanted into primates stimulated significant immune cell infiltration, immunoglobulin-G (IgG) binding, complement (C5b) and tumor necrosis factor-a deposition, and macrophage activation, while decellularized human (Alloderm^®^) and primate dermal matrices elicited a mild response [19]. This emphasizes the need for the removal of cells and cellular remnants from the ECM for clinical acceptance by the host [19].

Ingham and Fisher envisioned that decellularization of the tissue leaving an intact extracellular matrix (ECM) and reseeding the ECM would provide satisfactory implants. They evaluated the development of antibodies in response to donor allograft valve implant in patients who received cellularized and decellularized allografts. Decellularized grafts elicited lower levels of anti-HLA class I and II antibody formation after implantation than cellularized allografts, with patients in the latter group presenting donor-specific antibodies class I and II within 3 months of the observation period. They concluded that sodium dodecyl sulfate decellularization process is the best alternative to decrease the immunogenicity of allograft valve transplant [20]. In a two-part series, they evaluated the potential of developing a tissue-engineered heart valve containing autologous cells, enabling the valve to maintain its biochemical and mechanical properties yet grow with the patient. The first study revealed that only sodium dodecyl sulfate (SDS, 0.03–1%) or sodium deoxycholate (0.5–2%) resulted in total decellularization at 24 h, while the major structural proteins had been retained and appeared to be intact based on histological analysis [21]. The second study investigated the effect of 0.03% (*w*/*v*) and 0.1% (*w*/*v*) SDS on the mechanical integrity of porcine aortic valve leaflets as well as whole porcine aortic roots also treated with 0.1% (*w*/*v*) SDS. They concluded that SDS treatment produced a more extensible tissue with equal strength compared with the fresh aortic valve and, in functionality experiments with SDS-treated whole aortic roots, showed complete valve leaflet competence under physiological pressures (120 mmHg) as well as physiological leaflet kinematics [22]. When the aortic root wall was additionally thinned with 1.25% (*w*/*v*) trypsin treatment, then evaluated in vitro, decellularized intact porcine aortic roots presented complete leaflet competence under systemic pressures, increased dilation and effective orifice areas, reduced pressure gradients, physiological leaflet kinematics, and reduced leaflet deformation [23]. They evaluated the biocompatibility and recellularization potential of the acellular porcine valve matrix using porcine fibroblasts and smooth muscle cells (SMC). No contact inhibition of growth, or changes in fibroblast or SMC morphology were observed indicating the biocompatibility of the matrix. The SMC migrated throughout the leaflet matrix over four weeks, but there was no fibroblast migration into the matrix [24]. They also investigated the potential for re-seeding an acellular porcine heart valve matrix using human mesenchymal progenitor cells (MPC), demonstrating, for the first time, that human MPC have the capacity to infiltrate an acellular porcine valve matrix under static conditions in vitro. Seeded cells penetrated the center of the acellular graft after 4 weeks to 2% of homograft cell density, and the cells had a similar phenotype to native valve interstitial cells (vimentin+, alpha-smooth muscle actin+, heavy chain myosin slow-, desmin-). However, the re-seeded cells also expressed osteogenic markers (alkaline phosphatase, osteonectin, and osteopontin) [25].

The above examples show decellularization was used to remove the antigens from animal tissue (primarily pig and sheep) and implant these in humans with success, i.e., xenografts. However, the anatomical difference of the biped, upright humans and the quadruped animals is reflected in the (a)symmetry of the leaflets, further complicating the xenograft heart valve procedure. For optimal coaptation of the leaflets and to prevent regurgitation, a xenograft valve needs to be constructed from size-matched leaflets. An additional wrinkle with xenografts is their regulation as class-3 medical devices in the USA requiring costly and prolonged clinical trials, while minimally manipulated allografts are treated as human tissue, requiring minimal approval. Equivalent requirements are applicable in Canada, European Union, and other jurisdictions.

The key concept in tissue engineering is to provide scaffolds that can replace the defective tissue. Studies listed above demonstrate that decellularization is an effective method in diminishing immune reactions against the graft tissue. One aspect of tissue engineering is to provide simple acellular tissues scaffolds; another aspect is to re-seed the scaffolds with the patient’s own tissue and allow them to grow in a laboratory environment. The second approach has been proven to be very costly and not necessarily as successful as the first approach. It appears that the body has the innate ability to recolonize the acellular scaffolds with the correct types of cells and regenerate itself (vide supra). In the context of regenerative medicine, decellularization technology aims to provide acellular biological scaffolds for functional tissue repair and replacement, regardless of whether the defect was created due to an accident, illness, or natural aging. The ability to employ these acellular scaffolds in the active young population suffering from sports injuries, in newborn and adults afflicted with congenital defects, or in the elderly suffering from natural decline are equally important. Acellular scaffolds have been provided for heart valves, pericardium, tendons, meniscus, dermis, and axon sheets, among others.

### 1.4. Development of Dermis Products

Derwin et al. evaluated the effects of tyramine-substituted hyaluronan (THA) into an extracellular matrix (ECM). They determined that low molecular weight (57 kDa) THA immobilized to fascia ECM had no effect on the inflammation response or remodeling, and the inflammatory response and remodeling of biomaterial implants depends on the location of implantation, and therefore our animal models need to be carefully chosen [26]. They hypothesized that immobilization of tyramine-substituted hyaluronan (THA) into an extracellular matrix (ECM) scaffold may be a strategy to promote an anti-inflammatory response to the ECM. In US Patent 8,080,260 they disclose impregnating the decellularized ECM with high molecular weight hyaluronan derivatives (>1000 kDa), then crosslinking these, thereby incorporating a crosslinked macromolecular network of hyaluronan that is interlocked within said derived extracellular matrix, where the ECM is fascia lata, dermis, small intestinal submucosa, or pericardium [27]. They characterized the host response to decellularized fascia to ECM- and TS-HA-treated fascia ECM both with or without crosslinking. They concluded, “Together, the results suggest that this particular preparation of TS-HA treatment (at the concentration, molecular weight, and tyramine substitution rate used here) adversely affects the host response and elastic mechanical properties of fascia ECM. Hence, the preparation of TS-HA treated fascia described in this study would likely not be beneficial as an augmentation device for the repair of rotator cuff tendon or other soft tissues” [28]. The slower degradation of the THA-coated implants could be used to advantage for the graft to maintain sufficient strength and integrity during healing.

In addition to the tissues mentioned above, one of the biomaterials in the tissue engineering realm is the porcine small intestinal submucosa (SIS), which is easily incorporated into host tissue and remodeled. This product has been extensively used in reconstructive surgery with satisfactory results. Some authors claimed adherence and viability of human cells seeded onto SIS demonstrated that commercially available SIS specimens contained porcine nuclear residues and was cytotoxic in vitro, and clinical use of SIS has resulted in localized inflammation, suggesting the material can cause an immunological response in vivo [29], while others have demonstrated its biocompatibility [30]. Others have suggested SIS can shift the biochemical balance in a wound from chronic to an acute state [31]. Valentin et al. have compared the degradation of ECM scaffolds composed of porcine small intestinal submucosa (SIS), either non-crosslinked or carbodiimide (CDI)-crosslinked. They found that the crosslinked material was resistant to macrophage-mediated degradation [32].

Decellularized, crosslinked porcine dermis (Permacol™) is another well-known tissue engineering product that has been cleared by the FDA (K992556) [33]. It has been implanted in various surgical procedures, including as a vascular patch [34,35], gastrointestinal surgery [36], congenital diaphragmatic hernia repair [37], rib wall construction [38], and many other different surgical procedures since being licensed for use in humans. However, a disadvantage of this material is its lack of complete resorption, and in an animal model of bladder augmentation caused micro-calcification and irregular detrusor regeneration.

Another decellularization technology, MatrACELL^®^, in particular as applied to the dermis, has been developed by LifeNet Health and commercialized under the trade name DermACELL^®^ [39]. This technology also claims low DNA content, high suture retention force, and ultimate load at failure. Cazzell et al. report on the efficacy and safety of DermACELL in a prospective, multicenter study of complex diabetic foot ulcers (DFUs). The 47/61 patients who completed the 16-week study, of which the deepest exposed tissue was predominantly bone (N = 45; 95.7%) and the others were tendon (N = 2; 4.3%), all achieved 100% granulation (mean time 4.0 weeks) with an average of 1.2 ± 0.4 applications of implant. The average wound area was 29.4 ± 22.1 cm^2^ (range 2.1–113.6) at the beginning and showed wound area reduction of 80.3% at 16 weeks. DFUs of 15 cm^2^ or smaller were substantially more likely to close than DFUs larger than 29 cm^2^ (*p* = 0.0008) over a 16-week duration [40].

Richetti et al. reviewed the use of commercially available graft material, both synthetic and tissue-derived, in the context of rotator cuff repair [41]. They lament the lack of prospective controlled studies but point to benefits of tissue-derived matrices that are remodeled in a different fashion than synthetic grafts. They point out that host response to graft material may be influenced by exposure to synovial fluid or mechanical load, or both. Leigh et al. investigated this phenomenon in the literature on all of these and they found that the shoulder repair and abdominal wall repairs exhibited similar host response to xenograft fascia lata at 7 days; however, at 28 days the response in the rat shoulder elicited a unique host response from that seen in the body wall. They emphasize the use of an appropriate model to evaluate the host response [42].

The importance of the extent of decellularization on type of remodeling is demonstrated by Brown et al. [43]. They evaluated this phenomenon in a rat abdominal wall defect model using autologous body wall tissue, acellular allogeneic rat body wall ECM, xenogeneic pig urinary bladder tissue, or acellular xenogeneic pig urinary bladder ECM, and found that acellular tissue predominantly elicited M2 type macrophages and resulted in constructive remodeling, while those with cellular components predominantly elicited M1 type proinflammatory macrophages resulting in dense connective tissue or scarring. In the same vein, it is important to know what forces the repaired area will experience. Schneeberger et al. report forces around 228 N and discuss the effect of different types of suturing method on the failure strength [44]. In a human cadaveric study, Barber et al. showed a 19% increase in failure load and fewer failures at the suture–tissue interface for supraspinatus repairs augmented with human dermis (GraftJacket^®^) compared with nonaugmented repairs, demonstrating the benefit of augmentation [45]. The benefit of augmentation was also demonstrated by Shea et al., who observed 40% reduced gap formation under cyclic loading in specimens augmented with non-crosslinked porcine dermis (Conexa) over controls (*p* < 0.05), as well as increased ultimate load to failure (429 ± 69 N vs. 335 ± 57 N; *p* < 0.05) [46]. The benefit of augmented rotator cuff repair using acellular human dermal matrix was reported by Barber et al. in a prospective, randomized study [47]. Comparing control (20 patients) and augmented (22 patients) after a mean follow up period of 24 months (range 12–38), they found American Shoulder and Elbow Surgeons (ASES) scores were significantly better in the augmented group (*p* = 0.035), as was the Constant score (*p* = 0.008). There were also more intact cuffs in the augmented group than in the control group (85% vs. 40%, *p* = 0.01) as determined by gadolinium-enhanced MRI evaluation. The importance and benefit of allograft augmented repairs, as measured by the retear rate is reported by Lee et al. [48]. In a prospective, single-blinded, randomized controlled trial with a long-term follow-up study, they report statistically significantly lower retear rates in the allograft group over the control group (9.1% vs. 38.1%, *p* = 0.034).

In addition to the DermACELL and GraftJacket already mentioned above, many other acellular human dermal matrices such as Alloderm (LifeCell), Allomax^®^ (C.R. Bard/Davol), and FlexHD^®^ (MTF Biologics), bovine derived collagen matrices, e.g., Matriderm^®^ (Dr. Suwelack AG), and composites (silicone layer and bovine biomaterial) like Integra (Integra Life Sciences) have been developed and commercialized.

Matriderm has a 510 k regulatory clearance in the USA as a collagen wound dressing. It consists of bovine dermis-derived collagen type I, III, V (≥95,8%, *w*/*w*) and bovine ligamentum nuchae-derived elastin (≥1.8%, *w*/*w*), which are not crosslinked [49]. A recent article describes its first clinical experience in the USA on complex extremity wounds at 11 application sites [50]. They report the mean time to healing in the patients treated only with MatriDerm as 49 days (range 7 to 84).

Overall, skin substitutes may be classified as autografts, allografts, xenografts, synthetics, and composites of natural and synthetic materials. Such a classification is illustrated by Foley et al., who reviewed different materials [51]. A technology assessment report lists 76 skin substitutes that were on the market in 2020 [52]. We refer the reader to this detailed article regarding the constitution (cellular, acellular, natural, synthetic, composite), clinical uses, and outcomes of these graft materials.

## 2. Materials and Methods

The literature encompassing basic science papers and clinical studies germane to the creation of a thorough review the history of soft tissue processing and the use of comparative processed soft tissue xenografts and allografts were selected. The development of DermaPure (Tissue Regenix, Universal City, TX, USA) and dCELL technology (Tissue Regenix, Universal City, TX, USA), its basic science, and clinical utilization for human orthoplastic reconstruction is highlighted by clinical vignettes.

## 3. Results

### 3.1. dCELL Technology–Hypothesis, Development, Key Differentiators in Process

As mentioned above, Eileen Ingham, John Fisher, and coworkers at the Institute of Medical and Biological Engineering (iMBE) at the University of Leeds in the UK have been pioneers in decellularization technology. Their vision was to start with animal/human tissue to be replaced, remove cells and immunogenic components, retain ECM structure and histoarchitecture, retain microscale biomechanical properties and function, and regenerate in vivo with recipient endogenous cells [53]. This is based on the hypothesis that scaffold architecture would generate micro-biomechanical stimuli to drive appropriate cell function. They aimed to remove cellular material, including glycosaminoglycans and DNA, in order to minimize the immune reaction to the implant material while preserving its structure and function. Some of their work on heart valves and other tissues has been cited above. Their decellularization technology is trademarked as dCELL Technology. In order to commercialize the intellectual property, it was spun off in 2006 as a company, and Tissue Regenix^®^ (York, UK) was born. TRX BioSurgery, a Tissue Regenix company, markets the acellular dermis, DermaPure, which has been used to treat numerous clinical applications, including chronic non-healing leg and foot ulcers.

The development of the technology for porcine aortic valves and patella tendon is described in US Patent 7,354,749 [54]. In broad strokes, the technology consists of bursting the cells in a mildly alkaline buffer solution containing proteolytic inhibitors, removal of the cellular component with another mildly alkaline buffer solution, removing the genetic material using DNase and or RNase, applying a cryoprotectant, and sterilizing the tissue using gamma radiation.

Human cryopreserved aortic and pulmonary valved conduits and their decellularized counterparts were compared using histology, immunohistochemistry, quantitation of total deoxyribose nucleic acid, collagen and glycosaminoglycan content, in vitro cytotoxicity assays, uniaxial tensile testing, and subcutaneous implantation in mice. The decellularized tissues were free of cells or cell remnants, and decellularized pulmonary valve arterial wall and leaflet tissues were mostly free of staining with major histocompatibility complex Class I, but there was some evidence of residual staining in isolated areas of the decellularized aortic wall and leaflet tissues. They claim >97% DNA was removed from all regions (arterial wall, muscle, leaflet, and junction) yet leaflets (pulmonary or aortic) seem to retain more than the other tissue and overall average. The collagen content of the tissues decellularization was not decreased, and the glycosaminoglycan content was reduced, the extent depending on the tissue type and location. Moderate changes in the maximum load to failure of the decellularized tissues were observed, with the direction of change and statistical significance depending on the type and location of tissue [55].

They also applied the idea of de- and recellularization to pericardial tissue, using hypotonic buffer, SDS in hypotonic buffer, and a nuclease solution. They found no whole cells or cell fragments and no statistically significant differences (*p* > 0.05) in the hydroxyproline (normal and denatured collagen) and glycosaminoglycan content or ultimate tensile strength of the tissue after decellularization. However, there was an increased extensibility when the tissue strips were cut parallel to the visualized collagen bundles (*p* = 0.005). The decellularized material was deemed biocompatible due to absence of contact or extract cytotoxicity when using human dermal fibroblasts and A549 cells [56]. They demonstrated the biocompatibility by further seeding the material with dermal fibroblasts as well as subcutaneous implantation into a mouse model for three months. In the mouse model the explanted decellularized scaffold was infiltrated with myofibroblasts, CD34+ cells, and macrophages, indicating a healthy repair process. In contrast, the fresh/frozen and glutaraldehyde-fixed pericardia were encapsulated with a thick layer of inflammatory cells and fibrous tissue. Another major difference was the negligible calcium content of the decellularized pericardia [57]. They also compared the biomechanical properties of fresh and decellularized bovine pericardia to those treated with different concentrations of glutaraldehyde (GA). They observed no significant differences in the mechanical properties of fresh and decellularized pericardia, but there was an overall tendency for 0.05% and 0.5% GA-treated pericardia to be stiffer than their untreated counterparts [58].

Subsequent research expanded the applicability of the dCELL process to other tissues such as tendons and dermis. Historically, cellular bone-patellar tendon-bone (BTB) allografts grafts used for anterior cruciate ligament (ACL) replacement showed delayed remodeling, and failure to integrate early. It was postulated that acellular BTB grafts would lead to faster remodeling, integration, and restoration of joint function. This theory was tested with a porcine super flexor tendon (pSFT) and evaluation of its mechanical properties in vitro. Detailed and extensive analyses of the viscoelastic parameters showed that antibiotics were preferable over peracetic acid (PAA) as a bioburden reduction step [59]. Herbert et al. used the decellularization technology on human patellar tendons (PT) and evaluated the mechanical properties [60]. When compared to non-decellularized cohort consisting of contralateral tendons from the same donors, they found no statistically significant differences between the two groups for elastic moduli for the toe region and linear region, transition point coordinates, and strain energy density for increasing strain. These data suggested that decellularization had no effect on the material properties of human PT grafts under quasistatic conditions. They also evaluated the porcine super flexor tendon (SFT) after decellularization, including steps of exposing tissue to 0.1% (*w*/*v*) sodium dodecyl sulfate, proteinase inhibitors, and nuclease solutions. Decellularization eliminated cells and cell remnants, reducing the residual total DNA to only 13 ng/mg (dry weight) from an initial value of 303 ng/mg (96% removal); immunohistochemistry showed retention of collagen type I and III and tenascin-C, and reduction but not elimination of α-Gal (galactose-α-1,3-galactose) epitope. They found decellularized SFT was biocompatible in vitro and in vivo following implantation in a mouse subcutaneous model for 12 weeks. They did not observe any statistically significant differences in ultimate tensile strength (UTS), failure strain, or Young’s modulus of the collagen phase of the native compared with decellularized tissue specimens [61]. These acellular BTB grafts were shown to maintain their strength, and in a proof-of-concept study in sheep, after six months the BTB–ACL replacement showed good results, with regeneration of the tendon and osteointegration in the bone tunnels.

Fisher and Ingham applied the decellularization technology to menisci as well. Units treated with 0.1% (*w*/*v*) SDS, protease inhibitors, nucleases, and 0.1% (*v*/*v*) peracetic acid were characterized. Histological, immunohistochemical and biochemical analyses of the decellularized tissue confirmed the retention of the major structural proteins, elimination of the major xenogeneic epitope, galactose-alpha-1,3-galactose, but revealed substantial (59.4%) loss of glycosaminoglycans. The material was biocompatible, as determined by using contact cytotoxicity and extract cytotoxicity tests [62]. Subsequently they examined the host response to the scaffold in galactosyltransferase knockout (GTKO) mice and the capacity of meniscal cells and fibroblasts to attach to and infiltrate the acellular scaffold in vitro. To assess the host response, female GTKO mice were used that were either not immunized or immunized by injecting packed porcine red blood cells into the peritoneum of the mice at days 0 and 14, and tissue was implanted on day 28. Fresh porcine medial meniscal samples left untreated and α-galactosidase-treated tissue served as negative control, and decellularized tissue were test samples in a subcutaneous implant model for 3 months. The cellular infiltrates in the explants were assessed by histology and characterized using monoclonal antibodies against: CD3, CD4, CD34, F4/80, and C3c. Cellular infiltrates compromised mononuclear phagocytes, CD34-positive cells, and non-labeled fibroblastic cells; T-lymphocytes were sparse in all explanted tissue types, and there was no evidence of C3c deposition. The potential of acellular porcine meniscal scaffold to support the attachment and infiltration of primary human dermal fibroblasts and primary porcine meniscal cells was assessed in vitro, and shown to support the attachment and infiltration of primary human fibroblasts and primary porcine meniscal cells. They concluded that acellular porcine meniscal tissue exhibited excellent immunocompatibility and potential for cellular regeneration in the longer term [63].

The bladder tissue is relatively thick (1–5 mm when not distended) making it difficult to decellularize and still maintain its normal mechanical properties and elasticity. The dCELL technology overcomes this issue by exploiting the compliance of the bladder which can expand to >15-fold its contracted volume. Immersing and filling the bladder (or other membranous sac) into mildly alkaline buffer solution containing a protease inhibitor inhibit autolysis (e.g., ethylene diamine tetraacetic acid (EDTA) and aprotinin) both distended it and reduced its thickness. To further the decellularization, subsequent steps include a mildly alkaline buffer containing an anionic detergent, such as sodium dodecyl sulphate (SDS) or sodium deoxycholate at low enough concentrations to maintain the structure of the biological material but enough to aid in decellularization, and using a solution containing either DNase, RNase, or both to eliminate nucleic acids and provide a tissue matrix of limited calcification potential. Finally, a cryoprotectant solution may be used to aid in storage. In this case, pig bladder was used since it is readily available, and its size, physical, and mechanical characteristics are similar to humans [29]. This process resulted in a large reduction in DNA content (from 2.8 ± 0.1 µg/mg dry weight to 0.1 ± 0.1 µg/mg) [64].

The applicability of the technology was tested on human amniotic membrane for future assessment as a surgical patch and a delivery system for epithelial cells. Decellularization consisted of treatment with 0.03% (*w*/*v*) SDS, hypotonic buffer, protease inhibitors, and nuclease; terminal sterilization was with peracetic acid (0.1% *v*/*v*). The authors reported complete removal of cellular components while the histoarchitecture remained intact; all major structural components of the matrix, including collagen type IV and I, laminin, and fibronectin, were retained, DNA content was diminished, and tissue was biocompatible in vitro and exhibited no adverse effects on cell morphology or viability [65]. Subsequently, they showed its biocompatibility in vivo by subcutaneous implantation into mice for 3 months [66].

In summary, the dCELL technology developed by Ingham and Fisher primarily consists of decellularizing the tissue by lysing the cells in hypotonic buffer, removing the cellular debris with 0.1% (*w*/*v*) sodium dodecyl sulfate (SDS) in hypotonic buffer, nuclease treatment using RNase and DNase, and incorporating protease inhibitors for protecting the ECM structure and antibiotics to eliminate or reduce bioburden. The inventors claim that the anionic detergent concentration (i.e., 1% SDS) used in other decellularization methods either destabilizes the protein interactions and/or solubilizes them, leading to degradation of extracellular matrix (ECM) proteins [54]. The dCELL technology avoids these high concentrations with SDS being in the range of 0.03–0.1%, and thereby prevents structural deterioration, especially in the presence of protease inhibitors such as EDTA and aprotinin. These mild conditions appear to yield the desired results.

Peracetic acid has been used in some of their recipes for sterilization but has been abandoned due to its derivative effect on tissue. Terminal sterilization by gamma irradiation is now favored as the method to insure sterility. The examples above demonstrate that it is versatile and applicable to many types of tissue.

### 3.2. dCELL Technology Applied to Dermis

The essential steps of the dCELL technology as applied to dermis are shown in Figure 1. The process is very thorough in decellularization, but at the same time it is very gentle.

Hogg et al. have thoroughly studied the properties of decellularized dermis [67]. They found that residual DNA content in samples from three different donors (≥3 samples each) was on the average 0.22% (range 0.17–0.30%) of the original cellular dermis. In other words, the dCELL technology with its decellularization steps and use of DNase effectively removes 99.80% of the DNA present on the incoming tissue. How does this compare with other marketed products? Using published data, it was found that DermaPure, dermis processed according to the dCELL technology, has substantially less residual DNA (Table 1). In another study using human dermis, it was found that DNA content was reduced by 99% while extracellular proteins were mostly preserved [68].

How does the dCELL Technology process affect the resulting tissue? In other words, what are the effects of lower concentrations of anionic detergents, presence of proteases inhibitors, and gentle processing parameters on the resulting tissue scaffold? These are illustrated by the biomechanical characteristics of the tissue and the histological analyses. The biotechnical testing shows that DermaPure is as strong as the 2× thicker market leading decellularized tissue graft with respect to ultimate tensile strength and has nearly the same tensile elastic modulus, and similar suture peak load and maximum burst pressure as tissue of equivalent thickness (Table 2). This enhanced increase in tensile strength has directly led to the successful application of DermaPure to reinforce tendon, ligament, and joint capsule reconstructions, and even the creation of “neotendons”. The reconstruction of tendon retaining structures (tendon retinaculum), which are vital to prevent tendon “bow stringing” and loss of the tendon-muscle units’ power, is well served as well.

DermaPure is offered in numerous configurations, one being DermaPure^®^ Meshed (Figure 2). In this process, slits are cut into the dermis that allows the tissue to cover a much larger area than when non-meshed and allows for egress of fluids. With an expansion factor of 1.8, a 7 × 10 cm (70 cm^2^) DermaPure Meshed (at a 3:1 ratio) can cover 126 cm^2^ [69]. Having a pre-meshed product helps overcome two critical clinical hurdles, being a more efficient surgical technique to cover large soft tissue defects that require ingrowth, as well as the pre-meshed product mitigating difficulties encountered with meshing thick soft tissue substrates using standard surgical skin graft meshing devices. Additionally, meshed product further improves the cost effectiveness (discussed elsewhere).

This product has only been available in the United States and in the United Kingdom. It is expected to become available in other jurisdictions as their diverse regulatory requirements are met.

To summarize, the use of an extremely low concentration of single anionic detergent (0.01% SDS) removes donor cells and cellular debris, nuclease treatment removes >99% DNA, and proteinase inhibitors retain the natural structure [21] and biomechanical properties of the tissue [23], yielding a biological scaffold that promotes remodeling. Finally, terminal sterilization by irradiation to provide a sterility assurance level (SAL) of 10^−6^ essentially eliminating all microbial pathogens [72]. These contrast with processes that use harsh fixatives and destructive processing which yield tissue with altered biomechanical properties [22,73], which may be prone to poor performance, instability, and material fatigue, and incomplete decellularization may induce a cellular response or reliance on matrix remodeling [74].

### 3.3. dCELL Technology–Increased Angiogenesis and Diminished Fibrosis

Bayat and colleagues evaluated the effect of using a decellularized dermal allograft (DCD) in a clinical study as part of a one-stage therapeutic strategy for recalcitrant leg ulcers [75]. As described above, the DCD was produced from split-thickness skin grafts harvested from cadaveric tissue donors with all epidermal and cellular components removed from the dermis. The evaluation period was 6 months, at the end of which wound surface area decreased by 87%, and 60% of patients being healed completely. The average wound size was 13.1 cm^2^ and the average wound age was 4.76 years. Wound healing was associated with an increased hemoglobin flux peaking at 6 weeks (*p* = 0.005), then returning to baseline or below depending on subgroup analysis of ulcers existing less or more than a year. They obtained histological evidence of host cell migration, proliferation, and conversion of chronic ulcers to wounds with characteristics resembling acute wound healing. They concluded that DCD provided significant improvement in treatment-resistant leg ulcers without the need for hospital admission.

Since one of the issues in wound healing is scar formation at the injury site, which may affect both the functionality and appearance, Bayat and colleagues hypothesized that that structural and biomechanical variation between biomaterials may induce differential scar formation after cutaneous injury and set out to evaluate the effects of wounds healed by secondary intention (control group), or treated by either autograft, decellularized dermis (DCD), or collagen-GAG scaffold (CG; Integra Matrix Wound Dressing; crosslinked bovine tendon collagen and glycosaminoglycan); the latter two lacked epidermal components of skin [76]. They evaluated the healing process weekly through 4 weeks with both the non-invasive method of optical coherence tomography (OCT) and biopsies examined through histology and immunochemistry. Additionally, biopsies were studied for changes in expression of genes and proteins via microarray analysis and polymerase chain reaction. They observed a thicker dermis for the DCD group at day-28 over the control group (*p* = 0.003) and CG group (*p* = 0.0001) approaching that of the autograft, lower collagen III levels than controls at day-7 (*p* = 0.032) and autografts at day-21 (*p* = 0.007), and sites treated with DCD uniquely exhibited late up-regulation of matrix metalloproteinases 1 and 3, oncostatin M, and interleukin-10 (*p* = 0.007, 0.04, 0.019, 0.019, respectively) (Figure 3). They suggested the late up-regulation of these genes not observed in controls may contribute to differences in the observed outcomes. They conclude that fibrogenesis was variable in skin substitute-treated wounds with reduced scarring in autograft and decellularized dermis (DCD) samples compared with controls, and that DCD may stimulate a more regenerative rather than reparative therapeutic option in cutaneous wound management. They suggest that structural and biomechanical similarities between DCD and autografts, i.e., native tissue, may contribute to the reduced fibrosis noted in their samples.

The chronic wound healing study described above showed increased hemoglobin flux and neovascularization, while the acute study highlighted the benefits of DCD over healing by secondary intent or a xenogeneic, man-made skin substitute collagen-GAG scaffold (CG, Integra Matrix Wound Dressing; Integra Life Sciences, Plainsboro, NJ, USA). They also evaluated the effect of DCD use on angiogenesis in acute wounds [77]. The study design is the same as the above acute wound study, except the evaluations were carried out using laser-doppler imaging and the study duration was 7 weeks. They observed increased hemoglobin flux compared to controls (*p* = 0.0035 at day-21 and *p* < 0.0001 at day-28) via full-field laser perfusion imaging (FLPI) and increased oxyhemoglobin concentration from day-14 onward via spectrophotometric intracutaneous analysis (SIAscopy) that was statistically significant (*p* = 0.0031, 0.0004, 0.0005, 0.0001 on days 14, 21, 28, and 42, respectively). The biopsy samples revealed vessel numbers derived from CD31-based immunohistochemistry were greater in DCD wounds at later time points (*p* = 0.046), which correlated with the statistically significant increases in mRNA expression of membrane-type 6 matrix metalloproteinase (MT6-MMP) and prokineticin 2 (PROK2). They observed that DCD samples showed a similar behavior to autografts, with granulation tissue formation at the margin between subcutaneous fat and DCD in the week after injury, and decellularized vascular channels reaching the host/graft interface providing access for native cells thereby allowing a rapid influx of endothelial and inflammatory cells. As seen with autografts, incremental infiltration of granulation tissue into these channels created a vascular network with red blood cells observed in multiple large vessels after 21 days. After 4 weeks, DCD closely resembled native tissue, with DCD colonized by migrating host fibroblasts and myofibroblasts and re-vascularized graft. In contrast, CG had expanded spaces between collagen fibers on day-7 that progressively matured, becoming fibrotic with reduced vessel density at day-21, with the CG implant remaining clearly recognizable on day-28. They concluded that treatment with DCD resulted in increased angiogenesis after wounding, and the significantly elevated mRNA expression of pro-angiogenic PROK2 and extracellular membrane-type 6 matrix metalloproteinase (MT6-MMP) seen only in this treatment group may have contributed to the observed responses. They point out that PROK2 enhanced endothelial cell proliferation and migration using wound healing assays, and MT-MMPs (particularly MT1-MMP) are critical regulators of endothelial cell invasion into 3-dimensional collagen or fibrin matrices and promote endothelial cell tubular morphogenesis.

The process steps described above appear to also influence the beneficial effect of DermaPure in acute and challenging chronic wounds. In a retrospective multicenter study, they found DermaPure healed all wounds with one application and had nearly 100% healing by 20 weeks for wounds that were larger and of a longer duration than comparable studies [78].

### 3.4. Tissue Engineering’s Impact on Patient Care and Clinical Outcomes

To date, the lead author has experience with the implantation of DermaPure in over 186 surgical settings. The author prefers the use of DermaPure allograft tissue due to the characteristics, handling, growth characteristics, and the uniform applicability to a wide range of musculoskeletal indications, making the product more desirable clinically and from a supply chain standpoint. With the extremely favorable growth characteristics, physical strength, and handling properties of DermaPure, the lead author has found significant clinical benefits in a wide range of surgical applications. Among these surgical indications are surface wounds, the reconstruction of loss of facial domain, reinforcement of tendons and creation of neotendons, reconstruction of tendon retinaculum, and notably for rapid healing of skin graft donor sites in multi morbid patients.

Surface Wounds: Diabetic foot ulcerations are our key indication for the use of DermaPure. This is especially true in difficult to heal areas, such as underneath the metatarsal heads and over pressure points exerted by bony structures. However, the use of DermaPure in traumatic wounds for patients who otherwise possessed medical risks deeming them surgically not candidates for skin grafting or the wound bed is of such condition that it mitigates placement of a skin graft, demonstrating the versatility of DermaPure for treatment of the most superficial portion of the soft tissue envelope (Figure 4).

A novel application for DermaPure developed by the lead author is the use of DermaPure to improve skin graft donor site healing in multi-morbid patients with poor skin healing, as well as to decrease skin graft donor site pain (Figure 5). An initial review (unpublished data) of 67 cases of DermaPure used as an adjunct to heal skin graft donor sites in multi-morbid patients resulted in a clinically significant decrease in the post-operative visual analog pain scale (VAS) (mean DermaPure VAS = 1.5 versus Xeroform control (author’s historical experience) mean VAS = 7.5). An observational meta-analysis of healing times in these multi morbid patients demonstrates a 25 to 50% improvement to healing of the skin graft donor site.

Loss of Fascial Domain: In situations of the loss of deep fascia, whether it be of the abdominal wall, the pelvic floor, regions of conjoined deep fascia of muscular attachments (i.e., the conjoined fashion of hip musculature with the fascia of the external oblique muscle), the sacral region, or deep fascial loss resulting in muscle herniation, DermaPure finds great utility in reconstructive surgery. Additionally, DermaPure has been found to be an excellent adjunct to provide a suspensory mechanism in breast surgery. All these applications enjoy success due to the rapid ingrowth and lack of inflammatory response within the surgical site associated with DermaPure.

Tendon Reconstruction: DermaPure has been found to have extreme value in the reconstruction of Achilles tendon injuries (Figure 6a).

Achilles tendon injuries with intercalary defects or after resection of diseased distal Achilles are particularly challenging to reconstruct, often requiring tendon transfer/grafts to restore function. An example is a long harvest of the flexor hallucis longus tendon (FHL) to reconstruct the severely injured Achilles tendon (Figure 6b).

However, sacrifice of the entire FHL tendon may result in gait disturbances where reconstruction of the FHL donor site is desirable. The lead author has developed a novel technique to reconstruct the FHL donor site after full length FHL harvest. This involves creating a “neotendon” composed of DermaPure which is tubularized to mimic the shape and gliding of the native FHL tendon, and is routed through the natural course of the FHL onto the distal great toes and tenodesed to the confluence of the lesser toe long flexors (Figure 7). This allows for the return of great toes flexion and grip strength required for balance and a normal gait cycle.

In the foot and ankle, the lead author has also successfully used DermaPure to reconstruct the superior perineal retinaculum, the extensor retinaculum at the ankle, the anterior and the metatarsal phalangeal joint capsules (Figure 8), and the knee retinaculum for exposed knee prosthesis followed by flap coverage (Figure 9). DermaPure may also be used to supplement padding when there is loss of the specialized fat pad of the plantar heel, such as after calcaneus fractures or other trauma.

In lower extremity amputations with loss of deep facial domain, DermaPure is useful to create a durable biologic fascia on which to rotate and secure muscle flaps and extend the domain deep fascia for padding over the tibial stump (Figure 10).

In the upper extremity, DermaPure has been used to reinforce tendon transfers, as well as reconstruct the tendon retaining retinacula (Figure 11). An additional indication for the use of DermaPure in trauma patients is the application over plates and screws in locations where the gliding of tendons over metallic implants place tendons at risk for attritional rupture.

DermaPure has also proven valuable in osseous procedures. When used in conjunction with the Masquelet technique of bone grafting, it provides a secure biologic chamber around antibiotic spacers as well as the final stage of bone grafting.

## 4. Discussion

This paper has thoroughly discussed the development and current specialized processing of human dermal tissue and the various clinical indications for DermaPure. The basic science of processing human skin substitutes for the use in human medicine is well established and sound. The favorable angiogenic profile and limited fibrotic response derived from dCELL processing technology favors the use of human skin over many other engineered products. The indications for DermaPure in orthoplastic surgery are quite broad, and the clinical outcomes are excellent. In over 185 cases where DermaPure has been utilized in orthoplastic surgery, the lead author has not observed any rejection of the implant, and a very low complication rate, even in multi-morbid patients. Incorporation is swift and reliable.

From a clinical perspective, the lead author has found a large number of practical clinical uses for DermaPure for reconstruction of the soft tissue envelope, as well as for the musculoskeletal system and specialized deep fascial tissues. DermaPure has proven to be reliable for multiple surgical specialties, including that of plastic and musculoskeletal reconstruction (ortho plastic reconstruction), with excellent predictable results and no appreciable adverse reactions or failures. Thus, from a clinical standpoint, we can recommend the use of DermaPure for a myriad of surgical needs. Indeed, the limitations of DermaPure for surgical reconstructions may only be limited to the creativity of the surgeon. Future advances may include the incorporation of human growth factors or signaling molecules specific to intended target tissue healing or regeneration.

## 5. Conclusions

The use of DermaPure has proven to be of great clinical value and the authors recommend its use as an augment to soft tissue reconstructions throughout the musculoskeletal and other integumentary systems, and soft tissue replacement for a wide variety of defects within a wide range of pathogenic processes/injury mechanisms. The potential uses of DermaPure in orthoplastic surgery is only limited to the creativity of the surgeon.

## Figures and Tables

**Figure 1 bioengineering-11-00422-f001:**
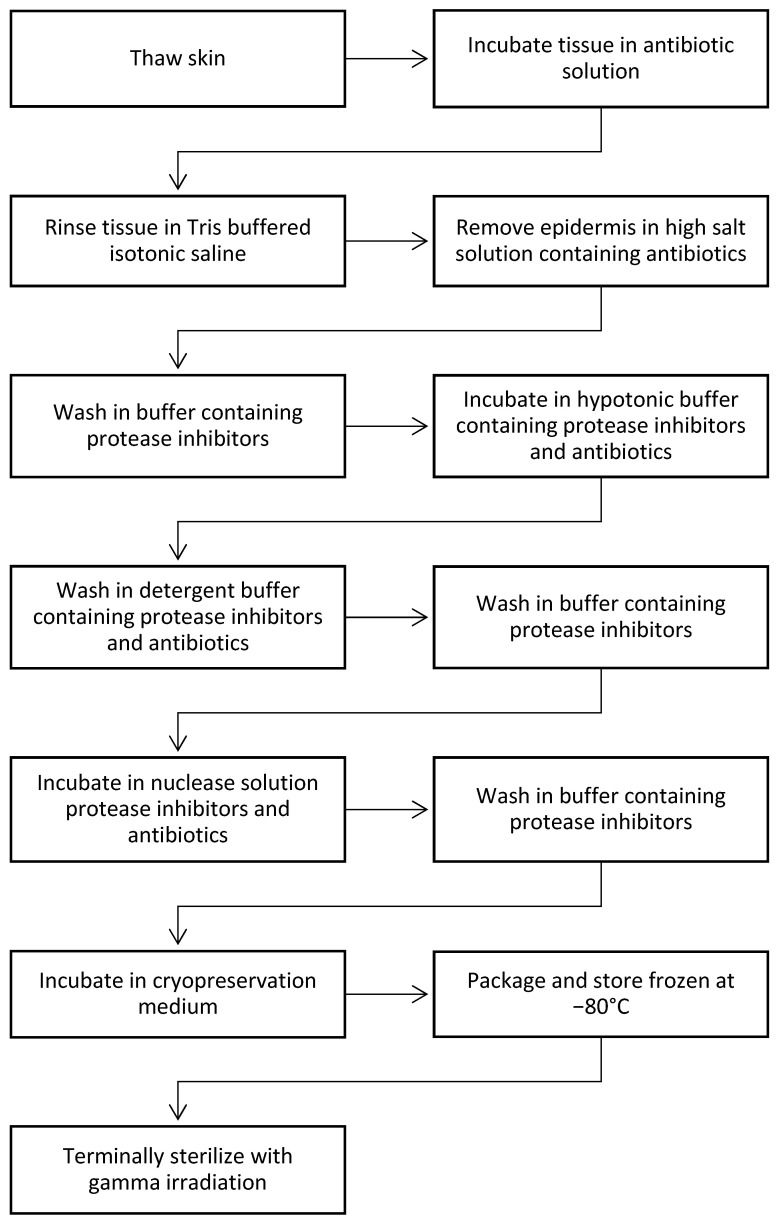
Steps used in dCELL process [67].

**Figure 2 bioengineering-11-00422-f002:**
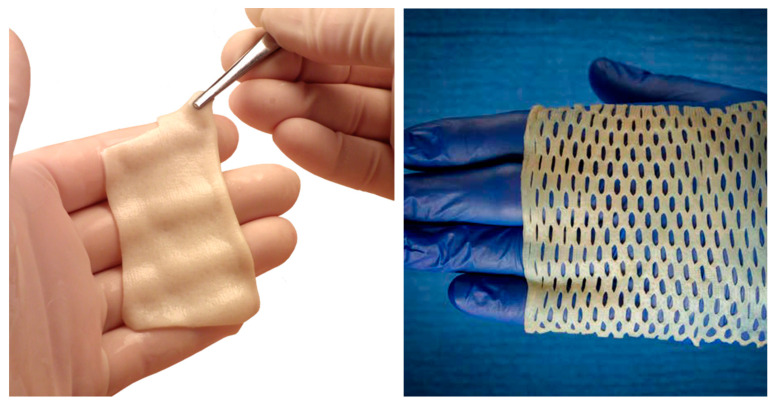
Photographs showing the DermaPure 7 × 10 cm (**left**) and DermaPure Meshed (3:1 expansion ratio) (**right**).

**Figure 3 bioengineering-11-00422-f003:**
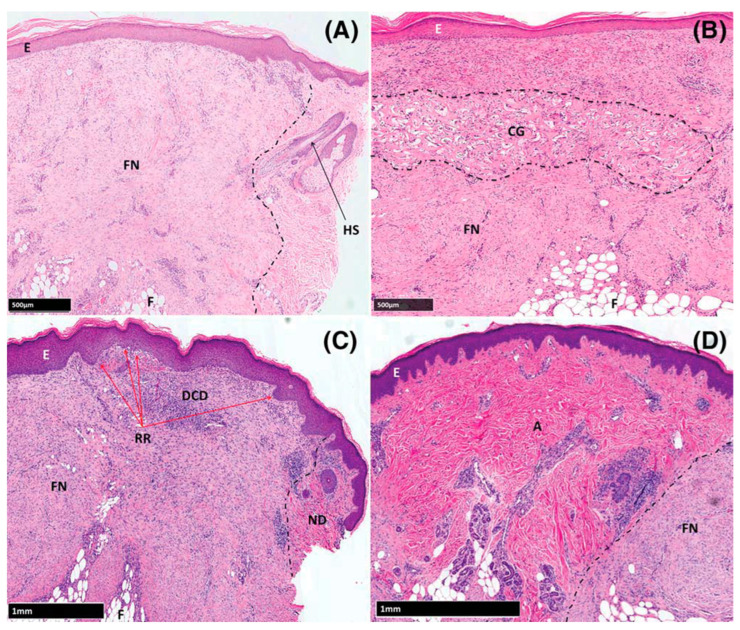
Haematoxylin and eosin-stained sections of formalin-fixed, paraffin-embedded tissue. (**A**) Control, (**B**) CG group, (**C**) DCD group, (**D**) Autograft group. Abbreviations: A: autograft, CG: collagen-GAG scaffold (Integra Matrix Wound Dressing), DCD: Decellularized Dermis (DermaPure^®^), E: Epidermis, FN: Fibrotic Neodermis, ND: Native Dermis, RR: Rete Ridges. Reproduced with permission from Greaves et al. [77].

**Figure 4 bioengineering-11-00422-f004:**
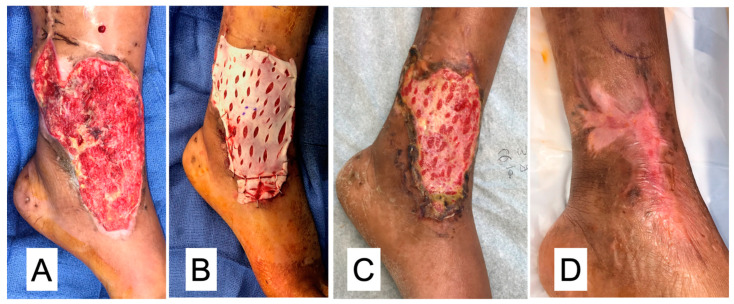
(**A**) Traumatic wound failing negative pressure wound therapy. (**B**) Application of DermaPure (fenestrated for drainage). (**C**) Incorporation at two weeks. (**D**) Complete healing at four weeks. Photographs courtesy of Dr. Christopher Bibbo, all rights reserved.

**Figure 5 bioengineering-11-00422-f005:**
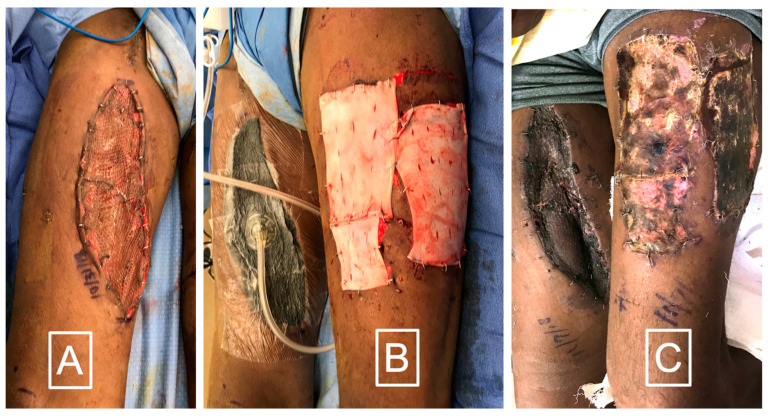
(**A**) Split thickness skin graft in a diabetic smoker after a necrotizing skin infection. (**B**) Immediate application of DermaPure to skin graft donor site. (**C**) After 10 days DermaPure has incorporated well. Skin graft donor site pain VAS was rated 1 on severity scale of 10. Photographs courtesy of Dr. Christopher Bibbo, all rights reserved.

**Figure 6 bioengineering-11-00422-f006:**
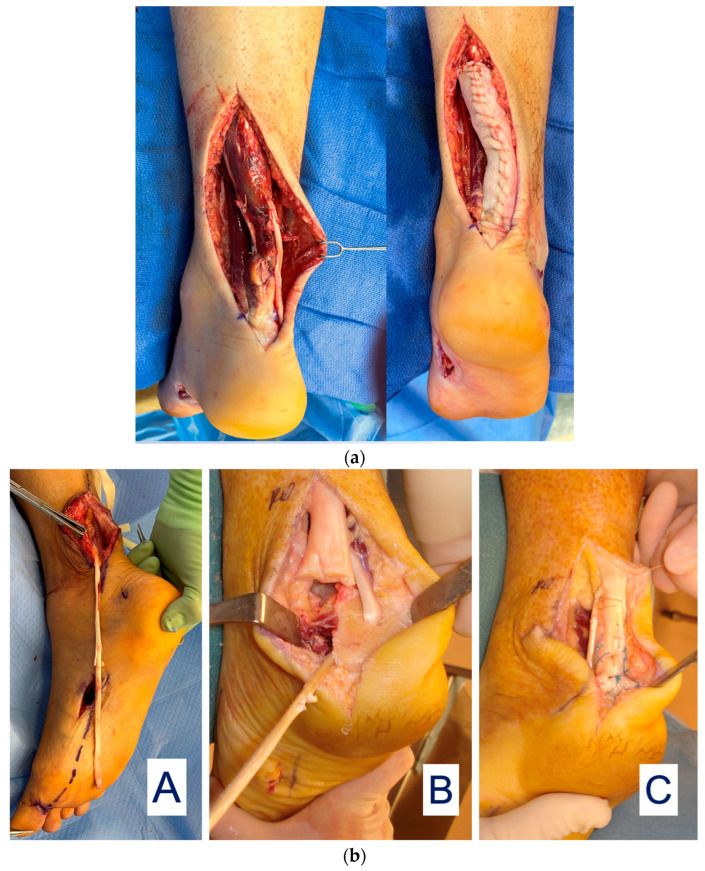
(**a**) Achilles tendon reconstruction (**left**) with DermaPure augmentation (**right**). (**b**) (**A**) FHL tendon harvested full-length; harvest of the required full FHL length which may result in loss of great toe grip strength and imbalance. (**B**) Long FHL tendon used to reconstruct distal Achilles defect. (**C**) Final reconstruction with FHL graft and re-insertion onto calcaneus. Photographs courtesy of Dr. Christopher Bibbo, all rights reserved.

**Figure 7 bioengineering-11-00422-f007:**
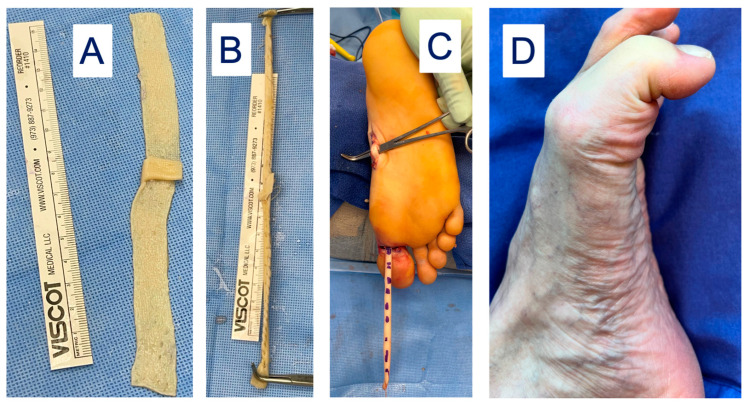
(**A**) DermaPure cut to same length of donated FHL tendon. (**B**) DermaPure tubularized to tendon shape (“neotendon”). (**C**) Neotendon composed of DermaPure (marked dashes) is routed through the foot and tenodesed to the long toe flexors in the midfoot, and the native insertion of the harvested FHL onto the distal hallux. (**D**) Final restoration of Hallux flexion through the metatarsophalangeal and interphalangeal joints, providing grip strength and gait stability. Photographs courtesy of Dr. Christopher Bibbo, all rights reserved.

**Figure 8 bioengineering-11-00422-f008:**
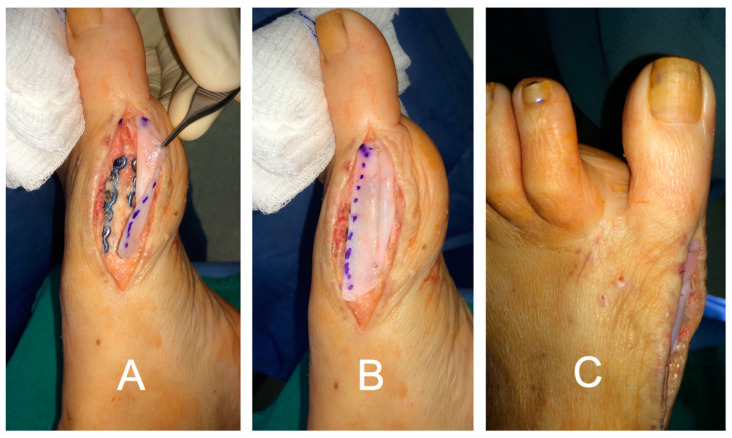
Great toe devoid of joint capsule. (**A**) DermaPure sutured to dorsal deep fascia and (**B**) plantar deep fascia to reconstruct the joint capsule. (**C**) Final repair with DermaPure stabilizing the joint and preventing pathologic toe deviation.

**Figure 9 bioengineering-11-00422-f009:**
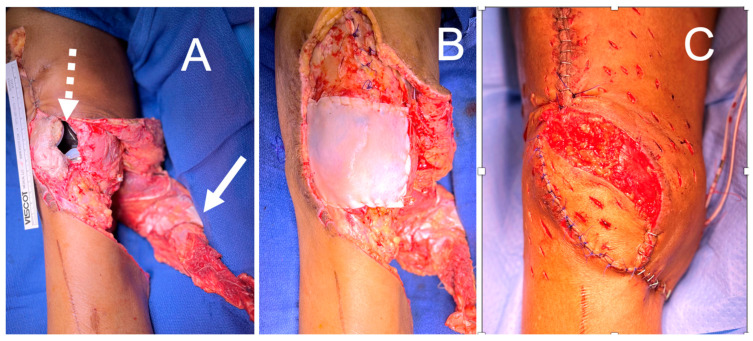
(**A**) Knee wound with exposed total knee arthroplasty (dashed arrow) and gastrocnemius muscle flap elevated for inset (solid arrow). (**B**) DermaPure reconstructed joint capsule. (**C**) Final reconstruction with gastrocnemius flap over DermaPure reconstructed joint capsule, retinaculum, and augmentation of extensor mechanism.

**Figure 10 bioengineering-11-00422-f010:**
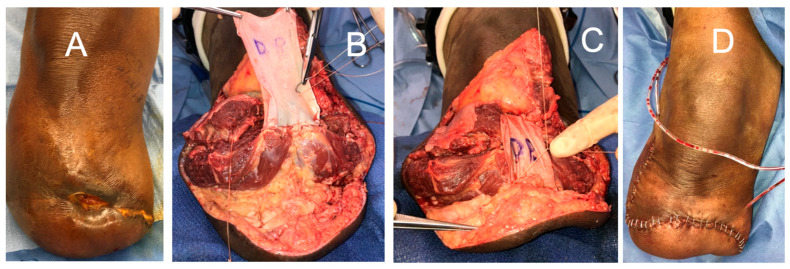
(**A**) Chronic wound in below knee amputation with loss of stump padding. (**B**,**C**) After “tri-muscle advancement flaps”, DermaPure sutured to gastrocnemius aponeurosis and deep leg fascia. (**D**) Final closure with vascularized muscle under skin margins and a robust distal stump pad. Photographs courtesy of Dr. Christopher Bibbo, all rights reserved.

**Figure 11 bioengineering-11-00422-f011:**
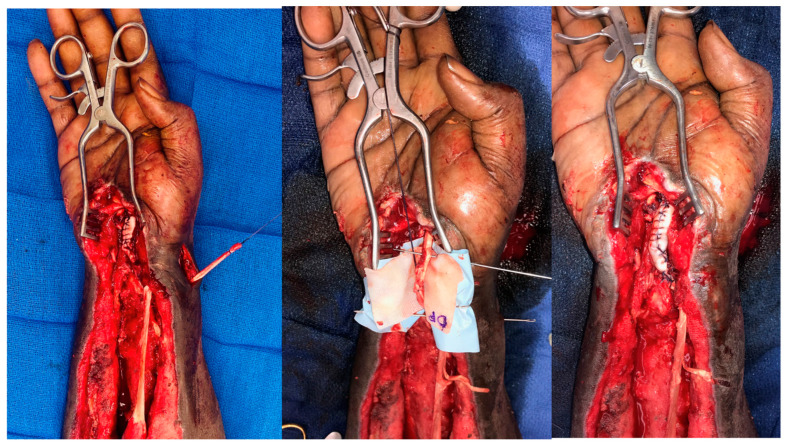
DermaPure augmentation of brachioradialis to flexor pollicis longus, and extensor carpi radialis longus to flexor digitorum longus transfers after necrotizing volar forearm infection. The patient regained thumb and finger flexion. Photographs courtesy of Dr. Christopher Bibbo, all rights reserved.

**Table 1 bioengineering-11-00422-t001:** Residual DNA content of commercially available decellularized dermis products ^1^.

Product	Residual DNA (ng/mg Dry Weight)	Fold Difference over DermaPure	References
DermaPure	1.26	N/A ^2^	[69]
MatrACELL ^3^	15.97	12.7	[39]
GraftJacket	134.66	106	[70]
Alloderm	272.80	217	[71]

^1^ The residual DNA findings in the table above were derived from studies that used the PicoGreen fluorescence assay to determine DNA content in decellularized tissue as compared to cellular tissue. ^2^ N/A: Not applicable. ^3^ Tissue processed with MatrACELL technology is marketed as DermACELL by LifeNet Health.

**Table 2 bioengineering-11-00422-t002:** Biomechanical properties of decellularized human dermis [69].

Biomechanical Test Parameter ^1^	Products			
	DermaPure	ML 1.0 mm	ML 1.5 mm	ML 2.0 mm
Sample Thickness (mm) ^2^	1.06–1.16	1.00–1.18	1.34–1.76	1.80–2.00
Ultimate Tensile Strength (MPa)	22.7 ± 5.7	16.4 ± 3.8 *	12.3 ± 3.1 *	22.6 ± 4.6
Tensile Elastic Modulus (MPa)	69.6 ± 6.7	52.8 ± 14.2 *	40.7 ± 6.6*	62.5 ± 5.6
Tensile Stiffness (N/mm)	18.7 ± 5.33	10.8 ± 2.7 *	12.8 ± 3.4	24.6 ± 5.8
Burst Maximum Load (N)	364.3 ± 63.5	368.4 ± 45.6	361.7 ± 69.4	682.5 ± 52.2 *
Burst Maximum Pressure (N/cm^2^)	1022 ± 398.7	1034 ± 128.0	1015 ± 194.8	1916 ± 327.6 *
Suture Peak Load (N)	50.0 ± 18.1	44.7 ± 7.22	44.7 ± 3.47	84.5 ± 11.7 *

^1^ Biomechanical Test Definitions: *Ultimate Tensile Strength (UTS):* Maximum resistance to failure and load carried by one square unit area (stress). *Tensile Elastic Modulus:* Measure of a material’s resistance to being deformed elastically (i.e., non-permanently) when a stress is applied to it. *Tensile Stiffness:* Measure of a material’s resistance to being deformed when a load is applied to it. *Burst Maximum Load and Pressure:* Measure of force and pressure required to rupture or puncture specimen. *Suture Peak Load:* Maximum force that can be applied before a suture pulls through the specimen. ^2^ Mean thickness range of samples measured prior to biomechanical testing. * Indicates statistically significant difference (*p* < 0.05) compared to DermaPure. Five samples of each thickness of tissue were used for burst, suture retention, and tensile strength tests. Results are presented as average ± standard deviation.

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
