# Peer review of "Decellularized Human Dermis for Orthoplastic Extremity Reconstruction"

_bioengineering, 2024, doi:10.3390/bioengineering11050422_

Round 1

Reviewer 1 Report

Comments and Suggestions for Authors

The manuscript is well-written and structured. I can only provide minor comments.

Can you include information regarding why decell animal products have not taken off compared to decell allografts?

Table 1. is not a table but rather a schematic. Please make this more graphical rather than simply a flow chart. you can include images depicting the processes to make the manuscript more readable/ accessible.

Comments on the Quality of English Language

Minor grammar changes should be checked through a native speaker / English center / or online application.

Author Response

  1. The manuscript is well-written and structured. I can only provide minor comments.

Can you include information regarding why decell animal products have not taken off compared to decell allografts?

Response:

This question is rather commercial in nature rather than a scientific or medical one.  Clinicians prefer the use of human tissue for allograft rather than animal products.  In addition, (a) historically xenograft tissue products have been associated with some forms of immunogenic response or inflammation due to the treatments used to address the immune response issues; (b) allografts are also more readily available in the U.S., and (c) animal tissue products need to go through a different regulatory approval process both in the USA and other jurisdictions. Decellularized animal tissue products are still a relatively newer technology so education and clinical experience will come over time.

At this time, we do not believe that discussion would improve the scientific contribution of the paper.

  1. Table 1. is not a table but rather a schematic. Please make this more graphical rather than simply a flow chart. you can include images depicting the processes to make the manuscript more readable/ accessible.

Response:

The reviewer is correct, and we have titled it as a figure, and renumbered accordingly.  There are no images related to processing is available to render it more graphical.

  1. Comments on the Quality of English Language. Minor grammar changes should be checked through a native speaker / English center / or online application.

Response:

Thank you, we have addressed the point.

Reviewer 2 Report

Comments and Suggestions for Authors

The manuscript is well written, complete in all information and above all very interesting with numerous in vitro tests and above all the authors show the clinical applications which seem to have truly interesting implications.

This reviewer has only two questions to ask the authors:

- can their product be used in combination with cells or stem cells? Whether to explain how the manuscript will be used

- is their product present on the international market? Is it possible to find it in Europe? If so how? What are the legislative aspects?

Please include these two points within your manuscript

Author Response

1   Can their product be used in combination with cells or stem cells? Whether to explain how the manuscript will be used

Response:

Theoretically the decellularized product could be repopulated with stem cells or the patient’s own cells as a form of tissue engineering.  However, no studies have been performed on this combination with this product.  The lead author has not used the product in combination with other cells or stem cells, thus cannot comment reliably upon this very interesting potential.

From a regulatory perspective, the product is accepted as a scaffold or used to augment soft tissue.  Providing repopulated product to the market would put it into a different regulatory category.  We are not privy to the company’s future intents.

2    Is their product present on the international market? Is it possible to find it in Europe? If so how? What are the legislative aspects?

Response:

The product has only been available in the United States and in the United Kingdom. As the reviewer noted, different jurisdictions have different regulatory requirements; some regulate this product as human tissue, some as a device or biologic with more extensive regulatory requirements.  This product may become more widely available as those requirements are met.  A sentence has been added to address this point, line 571.

Reviewer 3 Report

Comments and Suggestions for Authors

The authors present a review of decellularized human dermis development, characterization, and clinical outcomes of Dermapure.

The author seems to prefer and solely uses Dermapure in his clinical practice.

The article is missing a “conflicts of interest” section. Please clearly state if the authors have any financial or other conflicts of interest with the products named in this paper.

Section 1.3 and 1.4 – It would be interesting if the authors summarized these sections into a table with the different types of decellularized tissues and dermis products

Section 1.4 – Please add a paragraph on the use of Matriderm, a collagen-elastin-template, which serves as a dermal replacement scaffold

Section 1.4 – Please list all other products that are dermal substitutes (Alloderm, FlexHD, Integra, Allomax)

Line 593 – “wound” healing

Discussion – the authors should state the limitations of DermaPure (no product is without downfalls) and to state that further comparative studies with the gold standard (skin autografts and free flaps) are necessary

Author Response

  1. The author seems to prefer and solely uses Dermapure in his clinical practice.

Response:

We have added the following sentence, see line 682:  “the author prefers the use of DermaPure® allograft tissue to the characteristics and handling and growth characteristics and the uniform applicability to a wide range of musculoskeletal indications, making the product more desirable clinically and from a supply chain standpoint“.

  1. The article is missing a “conflicts of interest” section. Please clearly state if the authors have any financial or other conflicts of interest with the products named in this paper.

Response:

We thank the reviewer for pointing out our oversight.  We have addressed this point.

  1. Section 1.3 and 1.4 – It would be interesting if the authors summarized these sections into a table with the different types of decellularized tissues and dermis products.

Response:

Indeed, such a classification is interesting and has been recently provided in detail by Snyder et al. Rather than duplicating their work we have now referenced it along with an earlier one.

  1. Section 1.4 – Please add a paragraph on the use of Matriderm, a collagen-elastin-template, which serves as a dermal replacement scaffold.

Response:

A paragraph on Matriderm has been added.

  1. Section 1.4 – Please list all other products that are dermal substitutes (Alloderm, FlexHD, Integra, Allomax).

Response:

We have added a paragraph at the end of the said section, referencing an article listing names of 76 skin substitutes, their, types, clinical uses, and outcomes.

  1. Line 593 – “wound” healing:

Response:

We thank the reviewer; spelling error has been corrected.

  1. Discussion – the authors should state the limitations of DermaPure (no product is without downfalls) and to state that further comparative studies with the gold standard (skin autografts and free flaps) are necessary.

Rebuttal:

The lead author is extensively trained and experienced in microvascular surgery; Dermapure is not a substitute for any free flap, where multiple tissue composites are required, as well as the provision of continuous blood supply. Additionally, when keratinocytes are required, such as in a skin graft, it is also not a substitute. Thus, no changes are required.